# Seasonal Variability of Plankton Production Parameters as the Basis for the Formation of Organic Matter Flow in the Southeastern Part of the Baltic Sea

Sergey A. Mosharov [1,2,*], Irina V. Mosharova [1,2], Olga A. Dmitrieva [1,2,3], Anna S. Semenova [2,3]
and Marina O. Ulyanova [1,2]

1   Shirshov Institute of Oceanology, Russian Academy of Sciences, 117997 Moscow, Russia
2   Scientific and Educational Center "Geoecology and Marine Resource Management", Immanuel Kant Baltic Federal University, 236041 Kaliningrad, Russia
3   Atlantic Branch, Russian Federal Research Institute of Fisheries and Oceanography, 236022 Kaliningrad, Russia
*   Correspondence: sampost@list.ru

**Abstract:** The seasonal dynamics of production processes in the Baltic Sea are poorly studied. The aim of our research was to study the seasonal features of primary productivity (including the balance with bacterial production) and its redistribution in plankton in the southeastern part of the Baltic Sea in different seasons. More than 70% of primary production is formed in the 0–10 m layer (74–97% of the PP in the euphotic layer). In the same layer, PP accounted for almost 100% of the sum of primary and bacterial production in April and October, and almost 60% in June. Photosynthetic efficiency (PP/rETR) increased in June and October, demonstrating an increase in phytoplankton utilization of absorbed light energy. The depth-integrated values of PP, Chl *a*, bacterial, and phytoplankton biomasses were maximal in October. The maximum values of zooplankton biomass were determined in June, and they were significantly (5–14 times) higher than in other seasons. The maximum values of bacterial production were also in June.

**Keywords:** primary production; bacterial production; active fluorescence of chlorophyll; phytoplankton photosynthetic efficiency; phytoplankton; zooplankton; carbon polygon; Baltic Sea

## 1. Introduction

The Baltic Sea is one of the most studied seas in the world, but there is practically no information on the vertical distribution of phytoplankton and bacteria production processes in the Russian sector of the southeastern part of the sea. In general, studies of the production parameters of phytoplankton and bacteria in the Baltic Sea were carried out during the most productive period of the spring "bloom". The seasonal dynamics of production processes was mostly considered in the coastal areas of the sea [1–5].

Our research area is located in the Russian sector of the southeastern part of the Baltic Sea. Satellite data show that the study area is indirectly affected by the flow of the Vistula River. For example, in the article [6], the transport of suspended matter according to satellite images of the sea surface, which arrived with the river runoff of the Vistula into the Gulf of Gdansk, is considered using the example of one of the largest and most destructive floods over the past 100 years, which occurred in May 2010. It is shown that even with the inflow of significant volumes of river runoff (during the flood, it exceeded the average annual runoff by 6–7 times) under various hydrometeorological conditions, the influence of the Vistula River on the surface layer of the sea is minimal. In addition, the water area is under the influence of Major Baltic inflow events, which carry a volume of saline water from the North Sea [7]. The inflowing water in winter carries not only a higher salinity, a lower temperature, and higher oxygen content than the Baltic water, but also its specific

flora and fauna. As a result, in this area there is a mixture of flora and fauna of the study area, and species that enter with the transformed North Sea waters. The specific conditions of this area determined the choice of the location of the marine site of the Kaliningrad carbon polygon for the development and testing of carbon balance control technologies [8].

Phytoplankton-based food web is in many cases more efficient than bacteria-based pathways [9,10]. According to the "microbial loop" concept, dissolved organic matter (DOM) released by phytoplankton is consumed by heterotrophic bacteria that serve as food for protozoa. The latter serve as food for larger microplankton. The quality of bacteria as a food resource is lower than that of phytoplankton [10]. Thus, the ratio of autotrophic and heterotrophic production determines the efficiency of the food web and affects the entire trophic structure and productivity of the plankton community.

The seasonal dynamics of phytoplankton development in the Baltic Sea is characterized by three main biomass maxima leading to water blooms: spring, summer, and autumn. The spring maximum is formed in March–April and is mainly due to the mass vegetation of diatoms and dinoflagellates [11]. The summer maximum occurs in June–July, when cyanobacteria gain an advantage in development (*Nodularia spumigena*, *Aphanizomenon* spp., *Dolichospermum* spp.) [12]. The autumn maximum is observed, as a rule, in September–October, when the water temperature equalizes vertically as a result of mixing. During this period, diatoms dominate, for example, *Coscinodiscus granii* [13–15].

The ocean's ability to absorb $CO_2$ is primarily related to the "biological pump" (i.e., the sinking of photosynthetically produced organic matter into the deep ocean before remineralization occurs [16,17].

The biological pump is the incorporation of carbon into organic matter during photosynthesis and in the form of calcium carbonate by plant cells and its transfer from surface water to depths when immersed or eaten by vertically migrating animals [18,19]. Thus, the primary productivity of phytoplankton plays a substantial role in reducing the concentration of carbon dioxide in the surface layer, and hence removing it from the atmosphere [17]. After the death of planktonic organisms, all their organic and inorganic residues containing carbon settle into the bottom layer of the sea, but only a small part of organic carbon is fixed in bottom sediments [20].

The purpose of this work was to study the seasonal features of the vertical distribution of PP, BP, and the balance between them, as well as seasonal changes in the ratio of biomass of the three main components of the plankton community (phyto-, zoo-, and bacterioplankton), which determine the fate of particulate organic carbon (POC) in the marine environment, and accordingly, the formation of a downstream flow of POC.

## 2. Materials and Methods

### 2.1. Research Area and Sampling Procedure

The research was carried out in the southeastern part of the Baltic Sea on 28 April 2022, 28 June, 1 October, and 28 November 2021 (Figure 1). Water samples were collected from 0, 5, 10, 15, 20, to 25 m using Niskin bottles. The water samples were divided into subsamples, which were used for measurement of different parameters, such as chlorophyll concentrations, experimental carbon fixation estimations (PP), chlorophyll fluorescence, and abundance and biomass of plankton. Temperature and salinity profiles were obtained using a hydrophysical probe Sea & Sun CTD90M.

### 2.2. Measurement of Primary Production, Chlorophyll a and Light

Primary production (PP) was measured onboard using the $^{14}C$ uptake method [21]. The samples were incubated in polycarbonate flasks (50 mL) for 3 h in the original incubator with lighting and temperature maintenance. In the incubator, to simulate the light conditions corresponding to the sampling depths, each flask was illuminated by an individually adjustable LED panel (white light) with illumination level controlled using a LI-192SA quantum sensor. The light flux of each LED panel was regulated by changing the current.

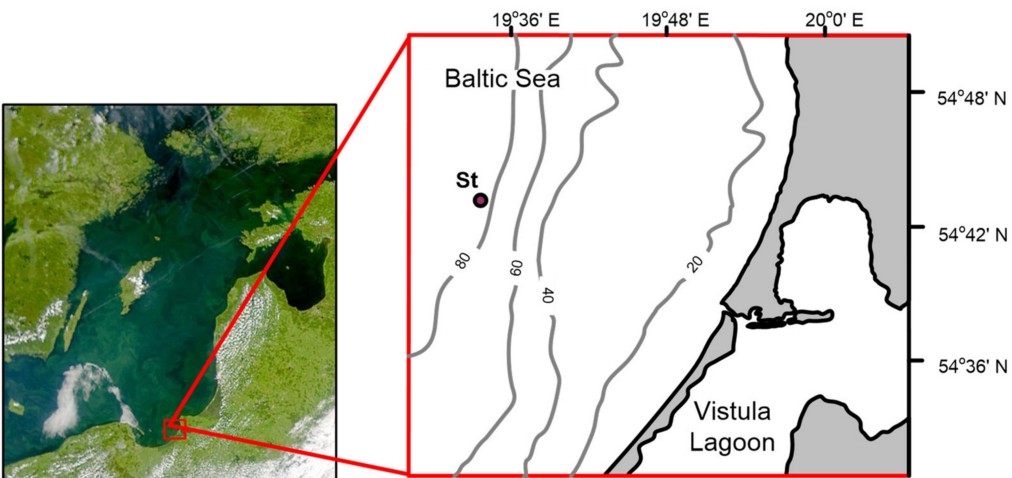

**Figure 1.** Research area and station location.

After incubation, the samples were filtered through a 0.45-μm filter (Vladipor, Russia). The samples radioactivity was determined using a Triathler liquid scintillation counter (Hidex, Turku, Finland).

The biomass-specific PP ($P^B$, mgC/mgChl *a* per day) was calculated by dividing the PP value by the concentration of chlorophyll *a* (Chl *a*) in the corresponding horizon.

Chl *a* concentration was determined by fluorometry [22]. Seawater samples (500 mL) were filtered onto GF/F filters (Whatman) at a vacuum of <100 mmHg. Filters were placed in tubes with acetone (90%) and stored at +4 °C in the darkness up to 24 h. The extracts was measured with a Turner Designs fluorometer (Trilogy fluorometer) before and after acidification with 1 N HCl. Calibration of the fluorometer was carried out before and after each cruise using standard Chl *a* (Sigma). The concentration of Chl *a* and phaeophytin a was calculated according to [23].

The underwater irradiance profile in the PAR range (photosynthetically active radiation, 400–700 nm) was measured using a complex including a quantum sensor Li-190R, an underwater sensor Li-193, and a Li-1400 DataLogger. The depth of the euphotic layer was determined from the depth of 1% surface PAR.

### 2.3. Measurement of Fluorescence Parameters

Active fluorescence of Chl *a* was measured with a fluorometer WATER-PAM (Walz). Prior to measurement, the samples were kept in the dark for at least 20 min [24]. The minimum ($F_0$) and maximum ($F_m$) fluorescence of the samples was measured. The maximum quantum efficiency of PSII ($F_v/F_m$) was calculated as [25]:

$$F_v/F_m = (F_m - F_0)/F_m. \tag{1}$$

As shown earlier, the maximum Fv/Fm values for phytoplankton under optimal conditions correspond to 0.70, with a significant difference between taxa [26,27]. The Fv/Fm value indicates the potential photosynthetic capacity of phytoplankton.

The value of rETR (relative rate of electron transport in PSII) gives an idea of the rate of flow of energy captured by chlorophyll into the zone of organic matter synthesis. The value of rETR at a specific light level is calculated as:

$$rETR = \Delta F/F_m' \times E \times 0.5, \tag{2}$$

where E is the actinic light level (μmol photons/$m^2$ per s), and the factor 0.5 is to correct for the partitioning of photons between PSI and PSII [28,29].

The effective quantum yield of photosystem II ($\Delta F/F_m{}'$) is measured as [30]:

$$\Delta F/F_m{}' = (F_m{}' - F_t)/F_m{}', \tag{3}$$

where $F_m{}'$ is the maximum fluorescence of light-adapted cells, $F_t$ is the fluorescence yield.

To determine $rETR_{max}$, $E_k$ water samples were exposed at 8 levels of light in the range from 0 to 1500 μmol photons/m$^2$ per s, including the natural light intensity at the sampling point and used in the incubator for $^{14}$C uptake measurements. A graph of rETR versus light level ("rapid light curve") was used to determine the optimal light level ($E_k$) for phytoplankton from a particular sampling depth, reflecting the level of light adaptation of phytoplankton.

### 2.4. Measurement of Abundance and Biomass of Phytoplankton

Collection and processing of phytoplankton samples were carried out according to standard methods [31]. Identification of phytoplankton species was carried out according to specific literature [32], etc. Clarification of the taxonomic status of individual species was carried out according to the algological database Algaebase [33]. Samples were concentrated by the sedimentation method. The laboratory processing of samples was carried out in the chamber "Uchinskaya" (volume 0.01 mL), according to standard methods using a Leica DM2500 microscope. Species that accounted for 10% or more of the total phytoplankton biomass were considered dominant. The content of organic carbon in the raw biomass of algae was taken equal to 10% [34].

### 2.5. Measurement of Abundance, Biomass and Ration of Zooplankton

Collection and processing of zooplankton samples were carried out according to standard methods [35,36]. Identification of zooplankton species was carried out according to specific literature [37,38], etc. To calculate the weight characteristics, the formulas for the length–mass dependence were used, or the figure of the organism was equated to a similar geometric figure [39–41]. The content of organic carbon in the raw biomass of zooplankton was taken equal to 10% [34,36,42].

For an approximate calculation of the relative value of daily rations of crustaceans, the formula $C = 0.0746 \times W^{0.80}$ [43] was used, which used the values obtained from a wet mass of individual organisms (W, g). Spending on the metabolism of organisms was defined as

$$R = (24Q_1 \times W_k \times 4.86)/c, \tag{4}$$

where ($Q_1 \times W_k$), $mlO_2/h$, the body oxygen consumption per unit of time; $Q_1$—a factor of metabolism at W = 1, K—constant, showing metabolism rate changes with increasing mass of the body. In the calculations we used values obtained [44] for crustaceans: $Q_1 = 0.125$ and K = 0.759; 4.86—oxygen–calories factor kal/$mlO_2$; c—calorie content of the organism, cal/g.

### 2.6. Measurement of Abundance, Biomass and Production of Bacterioplankton

To determine the number (NB) and biomass of bacterioplankton (BB), seawater samples were immediately fixed with 38% formaldehyde (preliminarily filtered through a filter with a pore diameter of 0.2 μm (Nucleopor)) to a final concentration of 1% in the sample and stored at 4 °C for a maximum of 24 h in the dark.

The number and size of bacterial cells were determined by epifluorescence microscopy using a fluorescent dye 3,6-bis(dimethylamino)acridine (acridine orange), and black nuclear filters with a pore diameter of 0.2 μm (Osmonics, Fort Lauderdale, USA) [45,46]. Preparations for microscopy were prepared using low fluorescent immersion oil (Olimpus, Tokyo, Japan). Bacteria were counted using a MikMed-3 LUM LED microscope connected to a Touptek Photonics FMA 050 digital camera (China) and a personal computer. The image was digitized using ToupView software and used for subsequent counting and measurement of bacterial cells. At least 200 cells were counted on each preparation and at least 50 bacterial cells were measured. The raw biomass of bacteria was calculated by

multiplying their number by the average cell volume. The carbon content in bacterial cells (C, fg C/cell) was calculated using the allometric equation most suitable for cells stained with acridine orange: C = 120 $V^{0.72}$ [46,47].

The growth rate and production of bacteriaplankton (PB) were determined in "live" seawater samples by changing their abundance by the "dilution" method. Water samples were deluted 1:10 with the same 0.2 μm-filtered water sample to eliminate bacteriotrophic organisms [48] and incubated in the dark at in situ temperature using temperature-controlled water baths for 24–48 h. The experiments were carried out in triplicate.

The specific growth rate of the number of bacteria (μ, $h^{-1}$) was calculated as:

$$\mu = (\ln N_t - \ln N_o)/t, \tag{5}$$

where $N_o$ and $N_t$ are the numbers of bacteria at the beginning and at the end of incubation, t is the incubation time of hours in diluted water samples.

PB (mg C/$m^3$ per day) was calculated as the product of the specific growth rate and the biomass (or number) of bacteria in undiluted seawater. The destruction of organic matter was calculated assuming that the ratio of bacterial production to their diet is 0.27 [49].

### 2.7. Statistical Analysis

Standard statistical methods of correlation analysis were used. Mean values are presented with standard deviation (±SD).

Depth-integrated values were calculated using the trapezoid method. The weighted mean values of the parameters for the water column were calculated by integrating the values over depth and dividing by the depth value.

## 3. Results

### 3.1. Research Area Hydrological Peculiarities

The depth of the euphotic layer in the study area on 28 April, 30 June, and 1 October, was 15 m at noon solar radiation of 2000, 500, and 1500 μmol photons/$m^2$ per s, respectively. On November 28, the euphotic layer was significantly deeper (25 m) with extremely low noon radiation (100 μmol photons/$m^2$ per s) due to dense cloudiness during this period.

In spring and summer, the upper mixed layer (UML) (water temperature 22 and 7 °C, respectively) was limited by a thermocline at a depth of 10 m. In early October, the thermocline was at a depth of 30 m with an upper water temperature of 15 °C. At the end of November, the water depth reached 60 m at a water temperature of 8.5 °C. Below the thermocline, the water temperature varied during the year within 4–8 °C. Salinity in all cases in the upper layer of the water column varied insignificantly from 7.2 to 7.5 psu. The halocline was located at a depth of 65 m throughout the year.

### 3.2. Chlorophyll a and Primary Production

The main part of the PP was in the upper 10 m layer (74–97% of the PP in the euphotic layer). It varied from 3.03 to 32.7 mgC/$m^3$ per day with a pronounced maximum in the 0–5 m layer in April (Figure 2a). In June, PP varied from 16.5 to 47.3 mgC/$m^3$ per day with a maximum at a depth of 5 m (Figure 2a). The bulk of Chl *a* was concentrated in the upper heated 10 m layer limited by the thermocline in April and June (1.83–2.96 and 4.6–10.7 mg/$m^3$, respectively). Below this layer, the Chl *a* concentration rapidly decreased with depth (to values of 0.3–0.4 mg/$m^3$).

In October, the maximum of PP was in the surface layer (147.2 mgC/$m^3$ per day) and was several times higher than the maximum values for the profile in other seasons (Figure 2a). In October and November the Chl *a* concentration was high in the entire 25 m upper unstratified layer (4.67–7.43 and 2.35–4.16 mg/$m^3$, respectively) with a maximum near the surface. As seen from Figure 2a, the PP values in the layer below 5 m were very close, and only in the surface layer did significant seasonal differences appear.

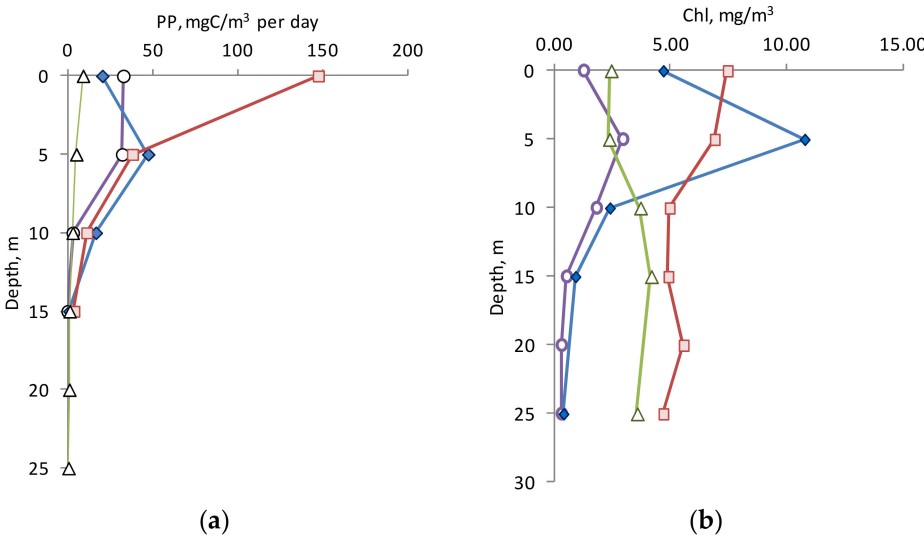

**Figure 2.** Vertical distribution of primary production (**a**) and Chl *a* (**b**) for April (circles), June (rhombus), October (squares), and November (triangle).

The maximum $P^B$ values were observed on surface waters in April and October (2.14 and 1.65 mgC/mg Chl *a* per h) and rapidly decreased with depth. In June, the maximum $P^B$ was at 10 m (0.58 mgC/mg Chl *a* per h). In November, the value of this parameter was low throughout the euphotic layer with a maximum at the surface (0.31 mgC/mg Chl *a* per h). Almost in all cases (with the exception of June), already at 10 m the $P^B$ value was low (0.07–0.19 mgC/mg Chl *a* per h).

*3.3. Photophysiology Parameters*

The quantum yield of PSII ($F_v/F_m$) within the 0–25 m layer was high in April, October, and November (0.543–0.814), and lower in June (0.327–0.585) (Figure 3a). The maximal $F_v/F_m$ value was detected in 10–25 m layer at all seasons.

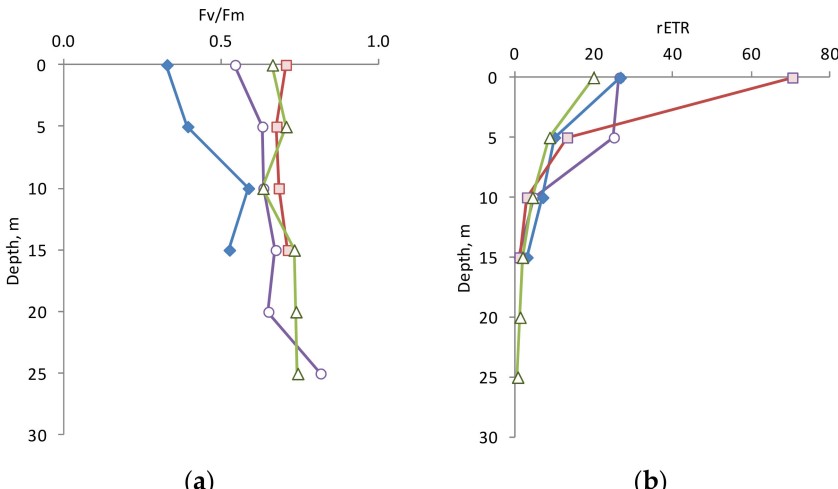

**Figure 3.** Vertical distribution of phytoplankton state parameters: (**a**) maximum quantum efficiency of PSII, $F_v/F_m$, (**b**) relative electron transport rate, rETR for April (circles), June (rhombus), October (squares), and November (triangle).

The rETR varied from 0.65 to 70.4 a.u. within the euphotic zone (Figure 3b) and decreased exponentially with depth as the light decreased. The surface rETR values were more changeable and varied from 19.9 to 70.4 a.u., whereas these values at 10 and 15 m were very close (3.0–7.0 a.u. and 0.9–3.0 a.u., respectively).

The relationship between the rETR and PP values in the euphotic zone at the same light levels showed a strong positive correlation (see the Discussion section). This allows us to use plots of rETR versus light level ("rapid light curves") to determine the optimal light level ($E_k$) for PP at sampling depth, reflecting the level of light adaptation of phytoplankton. Light saturation parameter ($E_k$) values derived from PAM rapid light curves were at the same level (710 μmol photons/m$^2$ per s) in the upper 25 m layer summer and autumn (Figure 4a). In the upper 10 m layer in spring and early winter, the value of $E_k$ was almost two times higher than in summer and autumn. It should be noted that in June, October, and November, the values of $E_k$ on the surface were the same (710 μmol photons/m$^2$ per s), while the average level of solar radiation for the light period varied significantly in different seasons (800, 250, and 50 μmol photons/m$^2$ per s, respectively).

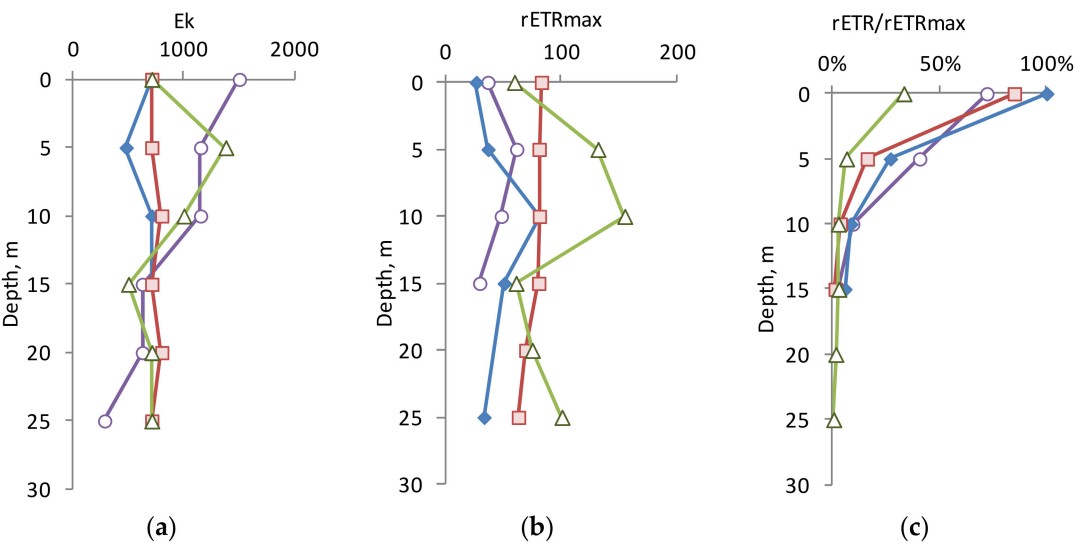

**Figure 4.** Vertical distribution of phytoplankton light adaptation parameters: (**a**) light saturation constant, $E_k$, μmol photons/m$^2$ per s, (**b**) maximum relative electron transport rate, rETR$_{max}$, a.u. (**c**) ratio of relative electron transport rate at ambient light (rETR) to rETR$_{max}$ (rETR/rETR$_{max}$) for April (circles), June (rhombus), October (squares), and November (triangle).

Generally, the phytoplankton in the 0–25 m layer was adapted to a high level of illumination. However, the maximum possible photosynthetic activity under optimal illumination (rETR$_{max}$) differed significantly in different periods (Figure 4b). In October, this value was constant (80 a.u.) in the upper layer, and in November it reached the maximum values for the vertical profile (131–154 a.u.) in the 5–10 m layer. In April and June, the rETR$_{max}$ values in layer 0–25 m were significantly lower.

The ratio of rETR/rETR$_{max}$ shows how the photosynthetic potential of phytoplankton is realized in real light conditions at a specific depth (Figure 4c). As can be seen from the figure, on the surface in April, June, and October, phytoplankton assimilate light energy to the maximum extent, while in November—only 33% of the maximum possible. At all seasons, this parameter decreased exponentially in depth, related to the decrease in underwater light, at all seasons.

### 3.4. Biomass and Production of Bacterioplankton

The BB depth distributions within the 25 m layer differed significantly in different seasons, and it apparently related to the UML thickness. In April and June, the UML was 10 m, and it was here that the maximum BB values were observed. The BB values significantly decreased in a deeper layer (Figure 5a). The same character of the BB vertical distribution, but with a smaller range of values, was observed in November. In October, the BB values showed a tendency to increase with depth. The highest BB values in the upper active layer (0–10 m) were in June and October (see Figure 5a) and exceeded similar

values for other seasons. Perhaps this is due to the higher water temperature (22.5 and 15 °C versus 7.2 and 8.5 °C, respectively). Based on the ratio of biomass to abundance in October, the cells were on average 1.4 times larger than in other seasons.

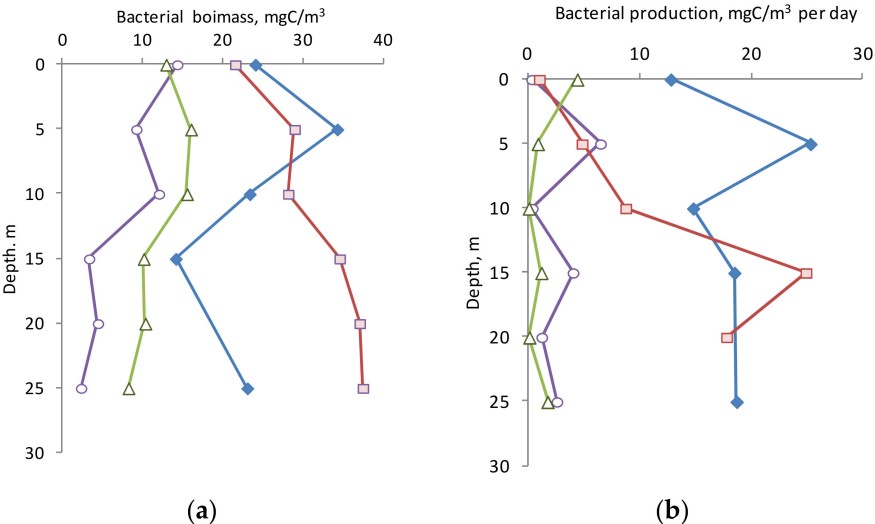

**Figure 5.** Vertical distribution of bacterial biomass (**a**) and bacterial production (**b**) for April (circles), June (rhombus), October (squares), and November (triangle).

PB values also varied significantly across seasons (Figure 5b). In June, bacterial production in the entire 0–25 m layer was high (on average $18.6 \pm 4.8$ mgC/m$^3$ per day), especially at a depth of 5 m. In October, the PB value in the upper 10 m layer was very low and increased significantly in the 10–25 m layer. In April and November, the BP values were at a low level and, on average, in the 0–25 m layer were $3.0 \pm 2.4$ and $1.3 \pm 1.6$ mgC/m$^3$ per day, respectively. The difference in PB levels by season, apparently, was also due to temperature conditions. The specific PB (P/B-coefficient) in April, June, and October varied greatly in the water column, but tended to increase with depth, while in late November it was vice versa.

*3.5. Biomass of Phytoplankton*

During the year, changes in the dominant (by biomass) taxa of phytoplankton were observed in the study area (layer 0–25 m): dinophyte algae in spring; dinophyte, cyanobacteria and cryptophyte algae in summer; diatoms and cryptophytes in autumn; and diatoms at the beginning of winter. The depth-integrated biomass of phytoplankton in the 0–25 m layer during the year increased from 1872 to 4694 mgC/m$^2$.

In spring, we observed 67 taxa of phytoplankton. Diatoms, dinophytes, and green algae were represented by the largest number of species. On average, for the 0–25 m layer, dinophytes dominated (66% of the total biomass). The dominant species in the communities were the following: diatoms *Chaetoceros* spp. (12–13% of the total biomass), *Thalassiosira* sp. (28%), cyanobacterium *Woronichinia compacta* (Lemmermann) Komárek and Hindák (13%), dinoflagellates *Gonyaulax triacantha* Jörgensen (19–34%), *Gymnodinium helveticum* Penard (11–12%), *Gyrodinium fusus* (Meunier) Akselman (14–15%), *Gyrodinium fusiforme* Kofoid and Swezy (11–19%), *Peridiniella catenata* (Levander) Balech (15–58%), *Protoperidinium brevipes* (Paulsen) Balech (16%), *Protoperidinium depressum* (Bail.) Balech (16–37%), a representative of green algae *Binuclearia lauterbornii* (Schmidle) Proshkina-Lavrenko (25%), and cercosa *Ebria tripartita* (J. Schumann) Lemmermann (13–14%). During this period, phytoplankton had a pronounced uneven vertical distribution in the water column. The minimum was noted at the 0.5 m (29.2 mgC/m$^3$), and the maximum at the 10 m (166.3 mgC/m$^3$). The average value of phytoplankton biomass in spring was $74.8 \pm 47.4$ mgC/m$^3$.

In summer, we observed 55 taxa of phytoplankton. Cyanobacteria, dinophytic, and green algae were represented by the largest number of species. For the 0–25 m layer,

cyanobacteria (31% of the total biomass), cryptophyte (16%), and dinophyte (43%) algae dominated in biomass. The communities were dominated by the diatom *Coscinodiscus granii* L.F. Gough (17% of the total biomass), the cryptomonad *Hemiselmis virescens* Droop (12–21%), and the cyanobacterium *Aphanizomenon* sp. (14–28%), dinophytes *Dinophysis acuminata* Claparède and Lachmann (18%), *Dinophysis acuta* Ehrenberg (19%), *Glenodinium danicum* Paulsen (10–20%), *Kryptoperidinium triquetrum* (Ehrenberg) Tillmann, Gottschling, Elbrächter, Kusber and Hoppenrath in Gottschling et al. (15–38%), *Prorocentrum cordatum* (Ostenfeld) J.D. Dodge (14–18%), *Scrippsiella acuminata* (Ehrenberg) (11%), *Chrysochromylina* sp. (13–14%), and cercosa *Ebria tripartita* (J. Schumann) Lemmermann (17%). The vertical distribution of the total phytoplankton biomass was characterized by a gradual decrease from the surface to depth. The maximum was determined in the surface (214.9 mgC/m$^3$), and the minimum at 25 m (17.6 mgC/m$^3$). The average value of phytoplankton biomass in summer was 89.7 $\pm$ 62.6 mgC/m$^3$.

In autumn, phytoplankton was represented by 51 taxa. Diatoms and dinophytes were represented by the largest number of species. For the 0–25 m layer, diatoms (76% of the total biomass) and cryptophytes (13%) algae dominated in biomass in October. The communities were dominated by the cyanobacterium *Microcystis aeruginosa* (Kutzing) Lemmermann (17% of the total biomass), diatoms *Cerataulina pelagica* (Cleve) Hendey (20–33%), *Coscinodiscus granii* L.F. Gough (18–48%), *Dactyliosolen fragilissimus* (Bergon) Hasle in Hasle and Syvertsen (19–33%), and cryptomonad *Plagioselmis prolonga* Butcher ex G. Novarino, I.A.N. Lucas and S. Morrall (11–15%). The phytoplankton biomass decreased from surface to depth. The maximum was noted in the surface (322 mgC/m$^3$), and the minimum at 10 m (102 mgC/m$^3$). The average value of phytoplankton biomass in autumn was 168 $\pm$ 70 mgC/m$^3$.

In late autumn, phytoplankton was represented by 47 taxa. Diatoms, dinophytes, and green algae were represented by the largest number of species. For the 0–25 m layer, diatoms dominated (98% of the total biomass)—one species of diatom *Coscinodiscus granii*. The vertical distribution of biomass was characterized by a relatively uniform distribution in the water column. The maximum was noted at 7.5 m (293 mgC/m$^3$), the minimum at 20 m (82 mgC/m$^3$). The average value of phytoplankton biomass in spring was 188 $\pm$ 66 mgC/m$^3$.

### 3.6. Biomass of Zooplankton

The maximum zooplankton biomass in all periods of research was in the 0–10 m layer. The depth-integrated zooplankton biomass in the upper 25 m layer reached a maximum in June (2266 mgC/m$^2$).

In the spring, zooplankton was represented by 17 species. In addition to Copepoda, a spring outbreak of small-sized Rotifera *Synchaeta baltica*, as well as Cladocera *Evadne nordmanni*, was noted. In general, Copepoda and Rotifera dominated the biomass for the 0–25 m layer, while the proportion of Cladocera was also high. The biomass of zooplankton decreased with depth (from 75.2 to 2.8 mgC/m$^3$). The average value of zooplankton biomass in the 0–25 m layer in spring was 23.4 $\pm$ 21.1 mgC/m$^3$.

During the summer period, zooplankton was represented by 21 species. During this period, large crustaceans *Bosmina (Eubosmina) coregoni*, *Centropages hamatus*, and *Temora longicornis* were dominated. In terms of biomass, Cladocera dominated in the surface, and Copepoda dominated in deeper layers. The total biomass decreased from maximum values in the 0–5 m layer (103–137 mgC/m$^3$) to a minimum at 25 m (58.3 mgC/m$^3$). The average value of zooplankton biomass in the 0–25 m layer in summer was 90.6 $\pm$ 30.0 mgC/m$^3$.

In autumn, zooplankton was represented by 18 species. *Evadne nordmanni* and *Acartia* spp. dominated. By biomass, in the surface and at 25 m, Copepoda also dominated, and in the zone of medium depths (5–20 m) Cladocera dominated. The proportion of Rotifera and Cladocera decreased with depth, while the proportion of Copepoda increased. The total biomass decreased from maximum values at 2.5 m (45.5 mgC/m$^3$) to a minimum

at 25 m (0.63 mgC/m$^3$). The average zooplankton biomass in the 0–25 m layer in autumn was 13.9 ± 12.1 mgC/m$^3$.

In late autumn, zooplankton was represented by 12 species, typical for the late autumn and winter period. Copepods, mainly represented by *Temora longicornis*, were dominated. The proportion of Rotifera and Cladocera decreased with depth, while the proportion of Copepoda increased. The biomass of zooplankton was low in the 0–25 m layer (3.92–10.4 mgC/m$^3$). The average value of zooplankton biomass in the 0–25 m layer at the end of autumn was 7.5 ± 2.5 mgC/m$^3$.

The diet of zooplankton with a filtration type of food increased from the spring period (5.63 mgC/m$^3$ per day) to the summer period (41.93 mgC/m$^3$ per day), when it reached its maximum values, then decreased towards autumn (4.67 mgC/m$^3$ per day), reaching minimum values at the end of the growth season (1.71 mgC/m$^3$ per day). Zooplankton exerted the greatest pressure on the phytoplankton community in the summer period, when, along with the maximum dietary values, it was dominated by large herbivorous crustaceans with a filtration type of food; in the spring period, small-sized rotifers dominated, which did not exert such pressure on the plankton community; the development of zooplankton and, accordingly, its diet also did not allow a significant impact on the phytoplankton community through grazing.

*3.7. Depth-Integrated Values*

The depth-integrated biological parameters (for euphotic layer) in different seasons are presented in Table 1. The integral values of PP, Chl *a*, bacterial, and phytoplankton biomasses were maximum in October. The maximum values of zooplankton biomass were determined in June, and they were significantly (5–14 times) higher than in other seasons. The maximum values of bacterial production were also in June.

**Table 1.** Depth-integrated values of chlorophyll *a* content (Chl *a*, mg/m$^2$), phytoplankton biomass (PhB, mgC/m$^2$), zooplankton biomass (ZB, mgC/m$^2$), bacterial biomass (BB, mgC/m$^2$), primary production rate (PP, mgC/m$^2$ per day), bacterial production (BP, mgC/m$^2$ per day), total (bacterial + primary) production (TP = PP + BP, mgC/m$^2$ per day), share of integral primary production of total production (PP/TP) in euphotic zone.

| Parameter | April | June | October | November |
|---|---|---|---|---|
| Chl *a* | 28.5 | 79.5 | 89.9 | 46.7 |
| PhB | 1310 | 1615 | 2589 | 2921 |
| ZB | 339 | 1719 | 243 | 120 |
| BB | 150 | 383 | 425 | 312 |
| PP | 257 | 375 | 633 | 75.4 |
| BP | 45.5 | 279 | 133 | 27.3 |
| TP | 302 | 654 | 766 | 102 |
| PP/TP | 85% | 57% | 82% | 73% |

The share of phytoplankton in the total biomass varied during the year from 43% to 90% (Figure 6). The maximum share was at the end of November. The proportion of zooplankton varied from 4% to 46% with a maximum at the end of June and a minimum at the end of November. The biomass of bacterioplankton during the whole year was an insignificant share in the total biomass of plankton (7–13%).

The variability in the ratio of the integral functional characteristics of the plankton community, reflecting the transformation of organic matter (OM) (PP, zooplankton feeding, and bacterial destruction), had a pronounced seasonality (Figure 7). The rate of PP exceeded the feeding rate of zooplankton bacterial destruction of OM in spring and autumn, and was significantly lower than them in summer. At the end of June, the intensity of consumption

of organic matter by zoo- and bacterioplankton was maximum, and several times higher than in other seasons.

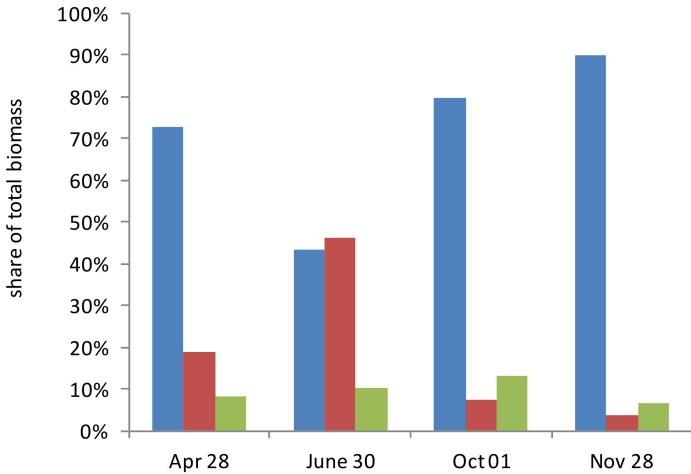

**Figure 6.** Changes in the share of biomass of phytoplankton (blue), zooplankton (red), and bacterioplankton (green) in the total plankton biomass in different periods of 2021.

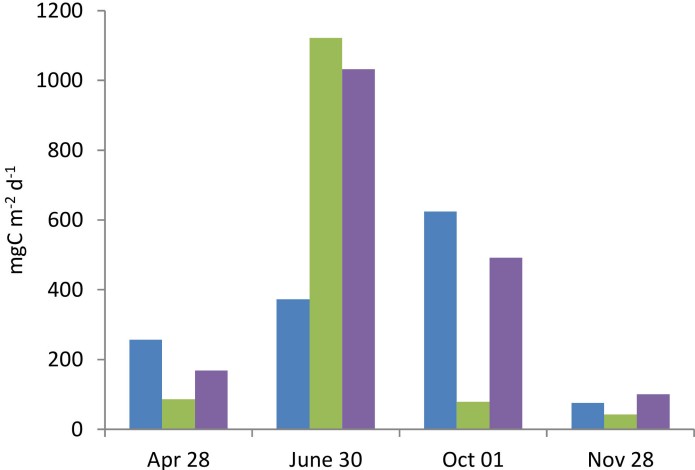

**Figure 7.** The ratio of the integral functional characteristics of the plankton community—primary production (blue), zooplankton ration (green), and bacterial carbon demand (violet)—in different periods of year.

## 4. Discussion

The dominant species of phytoplankton in June are characteristic representatives of summer phytoplankton complexes observed in the coastal zone of the southeastern Baltic Sea for a long period of time. Vegetation of *Dactyliosolen fragilissimus* and *Cerataulina pelagica* in October is not a typical phenomenon for the southeastern Baltic Sea. These species are characteristic of the more saline waters of the western part of the sea (the region of the Danish Straits, the Arkona Sea, and the Mecklenburg Bay). In November 2021, a monodominant community consisting of the diatom *Coscinodiscus granii* formed in phytoplankton, while the phytoplankton biomass remained at the level of October values of 188 mgC/m$^3$. *C. granii* is a typical representative of autumn phytoplankton communities in the Baltic Sea. In the autumn, it forms bloom, during which the phytoplankton biomass is characterized by uneven vertical distribution and reaches 3.5 gC/m$^3$ or more. At the end of the bloom, the number of algae decreases and their distribution becomes more uniform [50].

The integral PP value measured in our studies was comparable to the values obtained in the southern part of the Baltic Proper (381–617 mg/m$^2$ per day, [51]), but lower than in

the coastal zones of the Gdansk Basin—538–1214 mg/m$^2$ per day in the northeastern part of the basin [2,52] and 580–1860 mg/m$^2$ per day in the southwestern coastal zone [5]. The rETR values are comparable to those for the phytoplankton of the North Sea and the South Atlantic (2–40 a.u.) [53].

The potential photosynthetic activity of phytoplankton, determined by the maximum quantum efficiency of photosystem II ($F_v/F_m$), in the 0–25 m layer changed little with depth in April, October, and November and was quite high (0.534–0.742). In June, this parameter varied significantly in depth from the minimum values (0.327) on the surface to the maximum values (0.585) at 10 m. The high quantum efficiency of photosystem II of phytoplankton, i.e., the high potential activity of the primary (light) processes of photosynthesis in the upper 25 m layer, is also confirmed by the high maximum possible rate of electron transport in photosystem II (rETR$_{max}$) in this layer under optimal light.

The high potential of photosynthesis is realized in the spring, autumn, and pre-winter periods to the maximum extent only on the surface, where the P$^B$ value was almost 2–4 times greater than at a depth of 5 m. In June, the high potential activity of the photo-synthetic apparatus was at a depth of 10 m, which was expressed in the maximum value of P$^B$ for the vertical profile. However, the maximum PP value in June was at a depth of 5 m, which was due to the high content of Chl *a* at this depth.

A similar depth distribution of P$^B$ values and relatively high $F_v/F_m$ values was shown for various areas along the Antarctic Peninsula in late summer [54]. This indicates a high physiological plasticity for different phytoplankton.

The rETR values characterize the rate of conversion of solar energy into the chemical energy of the cell, which provides the processes of biosynthesis of OM by phytoplankton. They were at a maximum in the surface layer (0–10 m). The PP values in this layer were more variable than rETR (the difference between the minimum and maximum values was 48 and 23 times, respectively). Apparently, environmental factors have a lesser effect on the processes of absorbed energy than on the processes of its use in biosynthesis. We observed a similar situation in the Kara Sea, where PP was also more variable than rETR (14 and 8 times, respectively) [55].

A close relationship between the productive parameters considered in the work—rETR and PP—was observed in the upper 25 m layer (r = 0.91, *p* = 0.05, and n = 19). Linear correlations between ETR and total C fixation and/or O$_2$ production were established across regions [56–58]. We can consider these quantities as reflecting the intensity of the various stages of photosynthesis—light and dark, respectively [55].

For each particular day of measurements, an almost direct relationship was observed between the PP and rETR values at different depths. However, the slopes of the regression line differed significantly in different seasons (Figure 8). The slope values characterize the overall efficiency of using absorbed light energy in biosynthesis processes for a given phytoplankton. As seen from the figure, the maximum photosynthetic efficiency was in June and October, and the minimum in November. The maximum value exceeded the minimum by 7.8 times. We previously showed that a daily light decrease is significant for photosynthetic efficiency [55]. The established coefficients of photosynthetic efficiency can be used in the evaluation of PP according to active fluorescence measurements. Obviously, this coefficient is specific to each season [58]. Further research in this direction will improve the accuracy of PP estimates based on active fluorescence parameters.

According to our estimates, the BB values are well comparable with the values obtained earlier for the Polish part of the Gulf of Gdansk (30–70 mg C/m$^3$) [5], and are comparable with the values obtained in October 2007 in the Russian part of the Gulf of Gdansk (average BB—22 mgC/m$^3$) [52]. At the same time, the BB determined in our study in July was lower than the values obtained in July 2009: the average BB was 65 mgC/m$^3$ [52]. Apparently, this is due to the fact that our studies were carried out in more seaward waters than the coastal waters of the Curonian Spit and the Sambia Peninsula, where the cited work was performed. The PB values are consistent with the results obtained earlier for the Gulf of

Gdansk [5] and higher than the values obtained in the open part of the Baltic Sea (from 0.90 to 2.43 mg C/m$^3$ per day) [51].

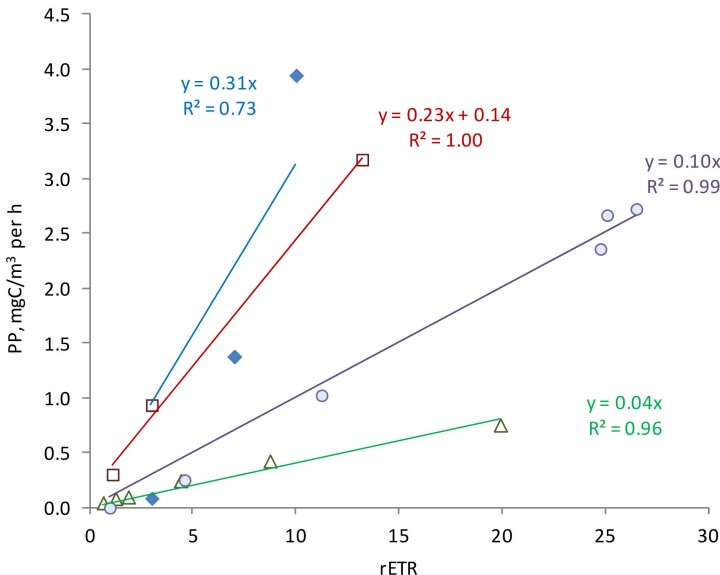

**Figure 8.** Relationship between primary production, PP, and relative electron transport rate, rETR, in the euphotic zone for April (circles), June (rhombus), October (squares), and November (triangle). The equation and determination coefficient of the regression are indicated on each regression line.

The total integral production of phytoplankton and bacteria was high in June and October, and several times lower in April and November (see Table 1). Throughout the year, PP accounted for a significant share of total production, from 57% in June to 85% in April. The share of PP in the basal production (sum of primary and bacterial production) in the upper part of the euphotic layer (0–10 m in April, June, and October, and 0–20 m in November) was 53–99% and decreased to 5–13% at the lower boundary of the euphotic layer, where PP limited to minimal light (Figure 9).

Similar studies carried out in estuary regions [1] showed that the production of heterotrophic bacteria is almost 100% of the total production in spring, while in summer the ratio of bacterial and primary production was approximately equal. In these areas, high levels of riverine DOC have a negative effect on PP (light attenuation) and a positive effect on BP (additional food source). In our studies, the share of BP in total production increased in summer, which may be due to the accumulation of dissolved and suspended OM (an additional food source for heterotrophic bacteria) in spring and early summer, as well as high water temperature (22 °C).

The ratio between the depth-integrated (for the euphotic layer) biomass values of the main components of the marine biocenosis (phyto-, zoo-, and bacterioplankton), which determine the formation of the POC flux, changed insignificantly during the year, while the dominance of the phytoplankton biomass remained; from 43% in June to 90% in November (see Figure 6). Zooplankton had a large biomass, comparable to phytoplankton only in summer, and in the rest of the year its biomass significantly decreased (by 5–14 times), and its share in the total biomass was from 4% in November to 19% in April. The share of bacterial biomass in the total plankton biomass during the year was small—7–13%. The most significant changes in the structure of the plankton community occurred in June, when a decrease in the share of phytoplankton and a significant increase in the share of zooplankton in the total biomass were observed. At the same time, the absolute value of the phytoplankton biomass in June was 1.2 times greater than in April, while the increase in zooplankton over this period was 5 times. Apparently, this situation reflects the long-term accumulation of zooplankton biomass after spring bloom.

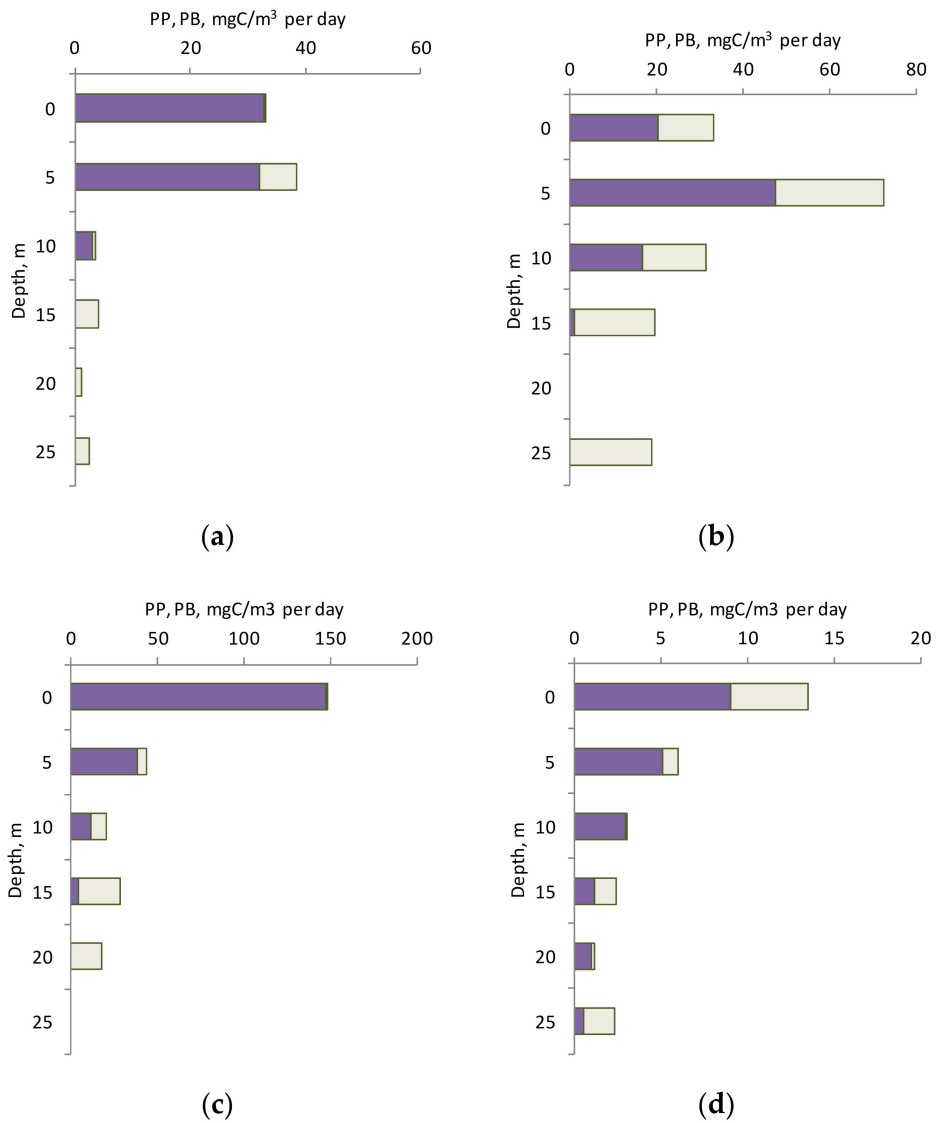

**Figure 9.** Vertical distribution of basal production and share of primary (dark) and bacterial (white) production in April (**a**), June (**b**), October (**c**), and November (**d**).

The main part (40–60% of the integral biomass) of all three groups of organisms in April was concentrated in the upper part (0–10 m) of the euphotic zone. In June, this vertical distribution generally remained. In October, the biomass of phyto- and bacterioplankton in the layer below the euphotic zone (15–25 m) was comparable to that in the upper layer (0–10 m). In November, when the euphotic zone occupied the entire 0–25 m layer, the vertical distribution of all three groups of planktonic organisms was almost uniform.

The functional characteristics of the biological components of the ecosystem related to the balance of POC in the marine environment (phytoplankton PP, zooplankton feeding intensity, and bacterial destruction), integrated for the 0–25 m layer, differed significantly during the year (see Figure 7). In June, the maximum activity of POC consumption was observed, which was more than two times higher than the intensity of photosynthesis. The high activity of zooplankton feeding in this period of the year was combined with the dominance of large herbivorous crustaceans with a filtration type of feeding. Obviously, zoo- and bacterioplankton consume additional POC brought in from outside or accumulated in the previous period. A similar distribution between the production and consumption of organic matter in the upper 25 m layer was also observed at the end of November, but at lower values. In April and October, the diet of zooplankton was lower than PP, and

bacterial destruction corresponded to the level of PP. High PP in October is combined with high bacterial carbon demand.

## 5. Conclusions

The main part of the PP at the marine site of the Kaliningrad carbon polygon was in the upper 10 m layer (74–97% of the PP in the euphotic zone). The maximum content of Chl *a* was in the upper heated 10 m layer in April–June and a uniform distribution in the 0–25 m in October–November.

The integral values of PP, Chl *a*, bacterial, and phytoplankton biomasses were maximum in October. The maximum values of zooplankton biomass were determined in June, and they were significantly (5–14 times) higher than in other seasons. The maximum values of bacterial production were also in June.

A close relationship between the productivity indicators—rETR and PP, which show the intensity of the light and dark stages of photosynthesis, were established. Photosynthetic efficiency (PP/rETR) increased in June and October, demonstrating an increase in phytoplankton utilization of absorbed light energy. The established coefficients of photosynthetic efficiency can be used in the evaluation of PP based on active fluorescence measurements. This coefficient is specific to each season.

The integral total production of phytoplankton and bacteria in the euphotic layer was high in June and October and several times lower in April and November. Throughout the year, PP accounted for a significant portion of total production, from 57% in June to 85% in April. PP constituted almost 100% of the total production in the 10 m layer in April and October, and about 60% in June.

The ratio between the integral (for the euphotic layer) biomass values of the main components of the marine biocenosis (phyto-, zoo-, and bacterioplankton), which determine the formation of the POC flux, changed insignificantly during the year, while the dominance of the phytoplankton biomass remained—from 43% in June up to 90% in November.

The functional characteristics of the biological components of the ecosystem of the marine site of the Kaliningrad carbon polygon related to the balance of POC in the marine environment (phytoplankton PP, zooplankton feeding intensity, and bacterial destruction), integrated for the 0–25 m layer, differed significantly during the year. Apparently, in summer, the POC synthesized by phytoplankton practically does not form a downward flow, but remains within the upper active layer of the water column in the form of biomass and metabolites of bacterio- and zooplankton. The feeding pressure of zooplankton on phytoplankton was maximum in summer, and it was determined both by the activity of feeding and the dominance of large crustaceans with a filtration type of feeding.

**Author Contributions:** Conceptualization, S.A.M. and M.O.U.; methodology, S.A.M. and I.V.M.; validation, S.A.M., I.V.M., O.A.D. and A.S.S.; formal analysis, S.A.M. and I.V.M.; investigation, S.A.M., I.V.M., O.A.D. and A.S.S.; writing —original draft preparation, S.A.M., I.V.M., O.A.D., A.S.S. and M.O.U.; writing—review and editing, S.A.M., I.V.M. and M.O.U.; visualization, S.A.M.; supervision, S.A.M., I.V.M. and M.O.U.; Project administration, S.A.M. and M.O.U. All authors have read and agreed to the published version of the manuscript.

**Funding:** Hydrological conditions analysis was supported by the state assignment FMWE-2021-0012 (Shirshov Institute of Oceanology, Russian Academy of Sciences). Biological investigations were supported by the state assignment FZWM-2021-0015 (Immanuel Kant Baltic Federal University) "Temporal variability of carbon fluxes at the carbon polygon in the Baltic Sea".

**Data Availability Statement:** The data are contained within the article.

**Conflicts of Interest:** The authors declare no conflict of interest.

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
