# Peer review of "Seasonal Variability of Plankton Production Parameters as the Basis for the Formation of Organic Matter Flow in the Southeastern Part of the Baltic Sea"

_water, doi:10.3390/w14244099_

Round 1
Reviewer 1 Report
The authors have presented an interesting study of plankton production in the southern Baltic Sea. The article has a logical layout with a detailed methodological part. Before publication, I suggest the following changes.
I do not fully understand the authors' intentions relating the results to two months side by side: October and November. I cannot agree with the statement that November is the beginning of winter (the calendar date is the third decade of December). Moreover, ice phenomena are recorded in this part of the Baltic Sea most often in January.
The authors point out that the analyzed area is under the influence of the Vistula River. Were the sampling dates correlated with the extent of this river's influence on the coastal zone of the sea? Specifically-what was the water flow on these dates? What was the wind direction and speed? This is a key piece of information for further detailed analysis done in the article.
In the Discussion section, reference should be made to many other interesting studies relating to the Baltic Sea. Example studies:
Reproductive Strategies of Non-Native Planktonic Crustaceans in the South-Eastern Baltic Sea,2017.
Revisiting Remane's concept: evidence for high plankton diversity and a protistan species maximum in the horohalinicum of the Baltic Sea, 2011
Biological diversity of plankton communities in the south-east of the Baltic Sea,2020
etc.
Reviewer 2 Report
The paper corresponds to the profile of the journal, contains interesting field research material and can be published after taking into account our recommendations.
1. It is desirable to redo the abstract, remove sentences that do not contain factual information. So, it is necessary to remove the first five sentences (lines 11-19), which do not contain fundamental information. The conclusion is well written, if it is shortened, it will turn out to be a good abstract.
2. In the introduction, the sentences on lines 67 – 78 should be moved to the beginning of the introduction. They are inappropriate in this part.
3. Write the objectives of the article more clearly.
4. In section 2.4, write which reference books were used to identify species. Also describe what equations were used to convert raw biomass into carbon units.
5. On lines 211 – 212, check the spelling of the sentence.
6. For ease of reading, it is desirable to give the units of measurement of parameters in Table 1.
7. It is desirable to give a reference on the line.
8. It is desirable to translate Figures 7-9 into the Results section, since they contain factual material.
